# Examining Physicians’ Approaches to Treating Relatives in Primary Health Care Centers: Insights from a Qualitative Study

**DOI:** 10.3390/healthcare12202021

**Published:** 2024-10-11

**Authors:** Manal R. Alhamdan, Nouf M. Aloudah, Saleh Alrajhi

**Affiliations:** 1Family Medicine Department, King Fahad Medical City, Riyadh 12231, Saudi Arabia; salrajhi@kfmc.med.sa; 2Clinical Pharmacy Department, King Saud University, Riyadh 11421, Saudi Arabia

**Keywords:** ethics, morality, conduct, qualitative, in-depth interviews, treating family members, theory underpinning, decision-making

## Abstract

**Introduction:** Family medicine physicians take care of a diverse population of patients with a variety of acute and chronic diseases. These patients include family, friends, and acquaintances who may ask for direct medical care or help in accessing healthcare products and services within or outside of officially approved procedures. This is ethically challenging due to an ambiguous medical code of ethics, but it is commonly accepted as normal behavior by society. The aim of this study was to explore family medicine physicians’ perspectives regarding the benefits, difficulties, and ethics of responding to medical care requests and/or favors from family, relatives, friends, and acquaintances and to make recommendations. **Methods:** The study sample consisted of junior and senior family medicine physicians working in primary healthcare centers affiliated with the Ministry of Health in Saudi Arabia. In-depth semi-structured interviews were conducted to collect data. Using social exchange theory, this qualitative study explores how family medicine physicians perceive and handle requests for medical favors from family members and others. **Results:** Semi-structured interviews were conducted with 19 family medicine physicians (six focus groups) with clinical experience ranging from 3 to 20 years. The data analysis identified three themes: perceived benefits and costs of cultural and social connectedness, shortcomings in patient management and healthcare systems, and recommendations to address challenges between physicians and patients who are relatives. **Discussion and Conclusions:** This study shows that treating others outside of normal access to healthcare services presents several ethical, moral, and professional challenges. Therefore, policy adaptation requires understanding this intricate dilemma and improving laws, system regulations, and guidelines for physicians and community members to improve access to care, reduce system abuse, empower providers, and enhance community awareness and compliance.

## 1. Introduction

In Saudi Arabia, family medicine physicians provide continuous and comprehensive healthcare for all individuals and their families, making them the most accessible physicians to the community [1]. They are in a unique position in their daily clinical practice, taking care of a diverse population of patients and managing a variety of acute and chronic diseases; they receive numerous requests for favors regarding medico-surgical care, pharmaceuticals, home medical equipment, and access to care.

Morality refers to the codes and expectations put forward by society for the conduct of individuals, groups, and organizations; they are derived from faith, religion, and established wisdom. It is a group of social rules accepted by the majority for interpersonal and intergroup social conduct and interaction, which govern general behaviors and actions towards the ethical issues faced by societies in medical and other areas of human living. In clinical settings, some ethical and moral issues are ambiguous, challenging, or cannot be resolved through commonly accepted ethical standards and moral codes of conduct [2,3]. For example, La Puma et al. found that 99% of physicians in a community hospital in the United States received requests from family members to prescribe medication (83%), to diagnose a medical illness (80%), and to help perform an emergency surgery (4%) [4]. Furthermore, Haneifi et al. found that all interviewed physicians in Malaysia received at least one request from family members during their careers [5].

According to the Saudi medical code of ethics, a healthcare practitioner is permitted to decline to treat a patient in non-emergency situations for personal or professional reasons that might compromise the quality of care. This is permissible as long as it does not harm the patient’s health and there is another qualified practitioner available to take over the patient’s treatment [6]. However, commonly accepted societal moral and ethical norms in Saudi Arabia encourage physicians to respond to personal and intrafamilial medical requests. Moreover, family medicine physicians in Saudi Arabia value the need to support strong relationships with their community members [3,7]. Thus, refusing care requests might negatively affect this relationship.

The American Medical Association Code of Ethics states: “It would not always be inappropriate to undertake self-treatment or treatment of immediate family members, in emergency settings or isolated settings where there is no other qualified physician available, physicians should not hesitate to treat themselves or family members until another physician becomes available” [8]. Therefore, treating oneself, family, and close friends is sometimes a gray zone in global ethics and practice [8]. The aim of this study is thus to explore family medicine physicians’ perspectives regarding the benefits, difficulties, and ethics of responding to medical care requests and/or favors from family, relatives, friends, and acquaintances and to make recommendations.

## 2. Methods

### 2.1. Study Design

This qualitative study employs an in-depth interviewing design to explore the perceptions of family medicine physicians regarding medical requests made by family members, friends, and acquaintances. This study is grounded in social exchange theory, which provides a theoretical framework for understanding the ethical determinants influencing physicians’ management of these requests.

### 2.2. Theory

Psychological and sociological theories provide a structured framework for understanding behavior, designing interventions, evaluating outcomes, and informing policy decisions. As a broader sociological theory, social exchange theory (SET) focuses on social interactions and relationships rather than on individual behavior change. Specifically, it seeks to explain social interactions and relationships in terms of the costs and benefits that individuals perceive in their interactions with others [9]. According to SET, individuals engage in social relationships and interactions because they believe that they will receive benefits greater than the costs involved [9]. SET has been applied to a wide range of social phenomena, including interpersonal relationships, organizational behavior, and social networks; it has been used to explain why individuals choose to engage in certain behaviors, such as helping others or risky behaviors, and to predict the outcomes of social interactions and relationships [9].

### 2.3. Participants and Settings

The Ministry of Health in Saudi Arabia is considered the main provider of healthcare services. Accordingly, the study sample consisted of junior and senior family medicine physicians from primary healthcare centers working for the Ministry of Health in Riyadh, Saudi Arabia. Purposive sampling was undertaken by emailing invitations to the primary healthcare centers for participation in the study. Interested participants who fulfilled the inclusion criteria were contacted to provide a telephone number to arrange an interview. For junior physicians, the inclusion criteria were physicians enrolled in the family medicine residency program under the Family Medicine Academy of the Ministry of Health with two to four years of work experience. For senior physicians, the inclusion criteria were physicians who were board-certified, held a consultant degree in family medicine, and were working full-time in the Ministry of Health with more than seven years of work experience.

Semi-structured interviews and focus groups were conducted face-to-face or virtually according to the preference of each participant. Face-to-face interviews were conducted at the participant’s workplace. The interviews lasted between 40 and 60 min. The interviews were conducted by the primary researcher; they were audio recorded and then transcribed verbatim. The interviews were conducted until data saturation was achieved, that is, when no new information emerged, according to Glaser and Strauss [10].

### 2.4. Topic Guide

The topic guide was created by the author (NA) based on previous literature [4,5,11,12]. Two physicians reviewed the topic guide for understanding and relevance to Saudi culture. The topic guide was used as general guidance and further probes were used to motivate physicians to express their point of view. The topic guide is presented in Appendix A.

### 2.5. Data Analysis

In the interview transcripts, the physicians’ names were replaced with corresponding pseudonyms. The transcripts were thematically analyzed using MAXQDA Analytics Pro 2020 (VERBI Software, Berlin, Germany). Thematic analysis is a qualitative research method used to identify, analyze, and interpret patterns or themes within a dataset. It involves systematically organizing and categorizing data to uncover meaningful insights and understand the underlying meanings and the experiences of the participants [13]. Each transcript was independently analyzed by two authors (M.R.A. and N.M.A.), and disagreements were resolved by discussion. As the semi-structured interviews progressed, data were analyzed after each interview to develop the initial codes and identify important and new emerging information. The final codes and themes were discussed with the third author (S.A.). Two researchers (M.R.A. and N.M.A.) were heavily involved in the data analysis (debriefing, coding, and interpretation).

In addition, to mitigate bias, each interview ended with a summary to be validated by the participants and to check for any ambiguity; after each interview, the two researchers (M.R.A. and N.M.A.) met and reflected on the information, came up with the main factors mentioned, and compared them with previous findings. Moreover, they maintained memos and journaling throughout the data collection and analysis using MAXQDA memos and online note software (Google. (n.d.). Google Keep. https://keep.google.com). The data from the interviews were kept on a locked, password-protected computer accessed only by the primary researcher. The data will be stored for up to five years after publication to enable the verification of data if challenged; then, all documents will be deleted and disposed of appropriately.

### 2.6. Ethical Approval

This study complies with the Declaration of Helsinki, and all of the participants provided informed consent for the recording and use of their anonymized responses. Ethical approval was obtained on 6 July 2022 from the King Fahad Medical City (KFMC) Research Ethical Committee. The IRB log number is 22-303. The IRB registration number with KACST KSA is H-01-R-012. The IRB registration number with OHRP/NIH USA is IRB00010471. The approval number of federal-wide assurance NIH, USA, is FWA0018774.

## 3. Results

The semi-structured interviews were conducted with 19 family medicine physicians with experience ranging from 3 to 20 years, practicing as residents (37%), senior registrars (37%), and consultants (26%). Six focus groups were held, each hosting 2–3 physicians. The participants’ characteristics are listed in Table 1.

The data analysis identified three themes: perceived benefits and costs of cultural and social connectedness, shortcomings in patient management and healthcare systems, and recommendations to address challenges between physicians and patients who are relatives. The next sections describe each theme and provide relevant quotations. Figure 1 illustrates the themes and subthemes.

### 3.1. Theme One: Exploring Perceived Benefits and Costs of Cultural and Social Connectedness

Participants shared their perspectives on the perceived benefits and costs associated with cultural and social connectedness when providing favors for relatives and friends. One aspect of the local culture that emerged was the value placed on interdependence, often referred to as “social connectedness”. The participants highlighted the profound impact of social support and deep connections with their families and others. Additionally, local values encouraging strong social support and networking were identified as significant factors that, at times, can influence physicians’ decision-making processes.

Furthermore, the participants discussed the use of practical wisdom by physicians during the decision-making process. Physicians engaged in discussions regarding specific issues they encountered and the ways in which they relied on their wisdom to make informed decisions. This exploration sheds light on the unique insights and experiences that physicians draw upon to navigate complex situations and apply practical wisdom in their professional practice. Figure 1 presents the challenges and rewards described by the physicians when treating family and friends. This theme is divided into two subthemes: challenges faced by doctors and physician rewards. Each subtheme is comprised of multiple components, which are presented below with supporting quotations.

#### 3.1.1. Challenges Faced by Doctors

##### Cannot Say No

The participants described their decisions as being affected by social influence in that their image would be affected if they refused to answer the request.


*“We are raised in a family-based community, where our life revolves around family, and good deeds are defined by helping and fulfilling different kind of requests, where you can’t actually say no, e.g., I will not help my grandfather’s friend. This is a bad stigma for the family and myself. So it’s difficult dealing with these scenarios in our community.”*
(Focus group 4—Participant 1)

##### Medicine Seen as a Humane Profession

The participants conveyed that embodying the role of the physician is synonymous with upholding high moral standards; therefore, declining requests for assistance could lead to the stigmatization of the individual as unhelpful within the community.


*“… the medical personnel are seen as [part of] a humane profession, and humane professions don’t have any rules and regulations; any human that needs help needs to be served no matter what, and that’s what causes confusion between what’s ethical and non-ethical.”*
(Focus group 3—Participant 1)

##### Exchange of Favors

The physicians highlighted that there exists a norm within the community, encapsulated by the principle of “if I help you, you help me.” Turning down requests for assistance could potentially hinder them from participating in this customary exchange of favors.


*“I respond to requests mainly to avoid hard feelings and benefit from them later on in their respected field.”*
(Focus group 1—Participant 1)

##### Fear of Blame

Furthermore, the physicians have a fear of being blamed for causing harm to the person making the request—a concern that compels them to decline the request.


*“If I treated my relative and it resulted as not expected, they will blame me, and even if they didn’t, I will somehow have self-blame. This is very difficult to deal with mentally, but if it is not a relative or a close person to me, I will not have this kind of self-blame because I have done my best and will not doubt myself.”*
(Focus group 7—Participant 1)

##### Obligation for Family Care

The physicians felt a strong sense of personal responsibility to help their family members in medical matters, especially when they were the only medical professionals in the family. This responsibility comes from a personal commitment rather than from community expectations, and family takes precedence in their view.


*“For me, the relatives whom I would help no matter what are only my mother and father, not even my siblings. Usually, for other relatives besides my parents, I would not talk to another doctor on their behalf.”*
(Focus group 1—Participant 3)

##### Emergency Cases/Do No Harm

The physicians agreed that a physician should provide consistent care, whether for a relative or any other patient. The focus is always on balancing the benefits against potential harm. If the potential for harm outweighs the benefit, the physician should step back and refer the patient to another competent physician.


*“It’s very unethical for a physician to prescribe medications to a family member or close friend, and in North America they may lose their medical licenses, but in urgent lifesaving, urgent care of an emergency case where if it’s not given it will result in negative outcomes to the patient, I think it’s a must to help, …”*
(Focus group 4—Participant 1)

##### Ethical and Legal Dilemmas

The physicians discussed that when faced with requests that cross ethical boundaries, a physician recognizes this as more than just a patient–doctor relationship issue—it becomes a matter of legality. The physician prefers to avoid such situations as much as possible.


*“When the request asked is not ethical, this relationship is beyond a patient and doctor, it’s a lawful act, I try to avoid such encounters as much as I can.”*
(Focus group 7—Participant 1)

##### Physicians Feeling Uneasy

The participants expressed discomfort with providing free consultations outside of work hours, even to family members. They would prefer if family members visited their clinic during working hours and followed the standard system before they felt comfortable providing medical care.


*“I don’t like to give a free consultation outside work hours even if it’s my family, but if they came to my clinic and followed the system, I would not refuse seeing them.”*
(Focus group 5—Participant 1)

#### 3.1.2. Physician Rewards

##### Satisfaction

According to the physicians, answering requests gives them a sense of personal fulfillment and a feeling of being valued and appreciated by those they serve.


*“I feel respected in the community because I’m a doctor and helping others gives me a feeling of self-satisfaction.”*
(Focus group 7—Participant 2)

##### Strengthening Social Networks

The participants described that fulfilling requests not only meets their immediate needs but also serves to deepen their relationships with those they assist.


*“First, strengthening the relation between me and my family members, when a relative asks for help it means he trusted me with his issue. Second, I personally feel confident and self-satisfied when helping a family member.”*
(Focus group 2—Participant 1)

##### Enhancing Knowledge

The physicians stated that they benefit from responding to the requests because they can enhance their knowledge by exploring various cases.


*“It also makes me research and read more when I get asked random questions, receive requests for favors by relatives, which will make me more confident in my daily practice.”*
(Focus group 2—Participant 3)

### 3.2. Theme Two: Exploring Shortcomings in Patient Management and Healthcare Systems

The second theme revolves around the shortcomings identified by physicians in patient management and healthcare systems, which impact their behavior. The physicians shed light on the challenges and limitations they face in effectively managing patient care within the current system as factors in providing (or not providing) favors to family and friends. The subthemes that emerged and the supporting quotations are provided below.

#### 3.2.1. Under/over Management

The physicians shared concerns about how treating family members might lead to bias in treatment and the possibility of influencing patient honesty, i.e., patients may communicate more openly with a doctor they do not know. Moreover, physicians are concerned about the risk of errors and the overcomplication of diagnoses.


*“This is called syndrome of the recommended patient, it’s when you have a patient you personally know and he may have something simple, but you complicate it and keep going in circles about it by overtreating or diagnosing.”*
(Focus group 1—Participant 1)

#### 3.2.2. Interfering with the Patient–Physician Relationship

The physicians discussed the topic of always respecting their colleagues’ treatment plans and the fact that they might undermine them, but they are sure that primary physicians might know more about their patients’ history. Physicians thus encourage their relatives to discuss any potential needs with their primary doctors. Furthermore, physicians might overlook certain aspects of care, and they encourage their relatives to be proactive in their discussions.


*“In cases where there is a need for an extra management or investigation, I will tell them just to ask their doctor if this is needed or not. Sometimes, as a doctor, we may forget, and a reminder will be appreciated. But without interfering with the management plan.”*
(Focus group 4—Participant 2)

#### 3.2.3. System Shortcomings: No Clear Guidelines

The physicians raised a concern about how some existing rules may lack logical justification, leading physicians or patients to circumvent the system to simplify processes for convenience. Furthermore, the physicians recognize that treating family members is a gray zone and suggest a case-by-case approach. They also acknowledge that they might treat family members but are aware that they will be held accountable for their actions. Moreover, the participants raised ethical questions about the fairness of withholding help when the system is difficult to navigate.


*“Sometimes some rules and regulations are not justifiable by logic, which usually lead to doctors or patients hijacking the system to ease things.”*
(Focus group 7—Participant 1)

### 3.3. Theme Three: Recommendations to Address Challenges between Physicians and Patients Who Are Relatives

The physicians offered recommendations to address the challenges that arise in the context of treating patients who are relatives. They provided insights and suggestions aimed at resolving these issues and improving the dynamics between physicians and their relatives. This theme presents a collection of practical recommendations that can help to inform interventions for better communication, establish clear boundaries, and enhance the overall quality of care in these unique patient–physician relationship dilemmas (Figure 1). The subthemes that emerged are presented below.

#### 3.3.1. Community Awareness

The physicians discussed the need to increase community awareness about health matters such as early screening, proper medication use, and healthy lifestyle choices. They recognized that a significant issue is the lack of knowledge about how to access healthcare services.


*“Advertising through mainstreaming regarding how to access medical care, how the medical system works, will relieve a lot of the pressure from our family and community.”*
(Focus group 4—Participant 1)

#### 3.3.2. Physician Training and Education

The physicians acknowledged a need for specific training to handle situations involving the treatment of relatives, which can be as complex as dealing with an angry patient. Such situations require professionalism to avoid conflicts of interest.


*“Residency training and building competencies to self-regulate, I think medical doctors may be the only career that is self-governed. You as a physician are responsible for your own actions and decisions, so raising awareness, competencies, and self-governing. Having a lot of rules and regulations may negatively affect the way healthcare is provided.”*
(Focus group 4—Participant 1)

#### 3.3.3. Guidelines

The physicians expressed a need for clear guidance on what is permissible when treating relatives in a clinical setting. They suggested that the rules may need to accommodate different contexts, such as treating a parent in a clinic versus at home, implying that some flexibility is necessary. They also highlighted that when new guidelines and regulations are established by policymakers, it is crucial to effectively communicate them to the intended audience to prevent misunderstandings and ensure compliance.


*“Having guidelines doesn’t mean being strict and restricted; it’s important in the long run to regulate the healthcare system. Also, having a flexible guideline is important for patient and physician satisfaction.”*
(Focus group 7—Participant 2)

## 4. Discussion

Our study explored family physicians’ views regarding treating family members and friends. Three themes were identified (Figure 1). We used social exchange theory to understand the physicians’ decision-making processes. The analysis showed alignment with the theory regarding how physicians see many perceived benefits when responding to the requests of family and friends, in addition to costs if they do answer the requests (Figure 1). Furthermore, the physicians highlighted additional factors that cannot solely be explained by social exchange theory, such as the shortcomings of patient management and healthcare systems that impact the decision to treat or not; for instance, requests from patients who do not have access to healthcare. The physicians suggested that community members were unaware of the risks of treating family and friends, and physicians need to be empowered by training and guidelines (Figure 1).

According to social exchange theory, individuals engage in social relationships and interactions because they believe that they will receive greater benefits than the costs involved [9], which was identified in the first theme of the perceived benefits and costs of cultural and social connectedness. The physicians discussed the costs and challenges faced, including the fact that they cannot say no to their relatives and friends for fear of stigma. Medical professions are also seen as humane professions, where it is not acceptable to refuse to help others. Additionally, physicians explained their fear of answering requests because it might cause blame, either from the requester or from themselves. However, physicians have also seen benefits in answering personal requests: they strengthen social networks with the requesters and increase their knowledge and awareness. Additionally, the physicians were satisfied and were acknowledged by the community.

Patient management and healthcare system shortcomings played a role in deciding whether to respond to the requests or not. The physicians stated that, in some cases, if they did not respond, the family or friend would not receive care. A study conducted in Saudi Arabia showed that the rapid increase in health expenditure in recent years caused a huge challenge; patient care for non-urgent cases may take months to years to be assessed by secondary/tertiary care providers, which results in patient dissatisfaction and disease progression [14]. This has a huge impact on family physicians working for the Ministry of Health and dealing with close family members and friends [14]. Also, a recent review of knowledge and attitudes regarding family medicine practice in Saudi Arabia found that 56.6% of the public did not understand the role of the family physician [7]. Even though the public trusts family physicians, they do not believe that they can treat certain chronic diseases such as diabetes and depression, which presents a huge problem in utilizing the new access to healthcare [7,15].

Moreover, the physicians in our study were holding back on answering the requests to avoid over- or under-management. When physicians treat family and close friends, their emotions can jeopardize their decision-making, and physicians might over- or under-treat [12]. Our study also showed that the physicians clearly did not want to interfere with the physician–patient relationship; physicians are aware of the consequences of responding to the requests but feel obligated to follow the no-harm rule [16]. The physicians further discussed the impact on patient autonomy [1], as well as interference with patient–physician relationships by providing medical advice and/or a management plan to a relative who is not their patient.

Furthermore, the physicians discussed the use of practical wisdom during the decision-making process; for example, they would follow the no-harm rule, and they would intervene if they were consulted by family and friends for an emergency or urgent issue, even if it did not allow them to avoid harm. Given the challenges in accessing healthcare [14], patients end up asking for help from a medical professional relative or going to private institutions. A study conducted in Mexico found that many households tend to pay privately for medical treatment, even when governmental insurance is available, and the reasons include the shortage of medications and the non-coverage of certain illnesses—a result of the healthcare system struggling to adequately provide healthcare access [17].

Our study showed that physicians are pushed to treat family members because they feel responsibility and obligation towards their family, close friends, and relatives. This obligation emerges from their own moral beliefs and from community pressure. Similarly, a qualitative study conducted with 22 primary care physicians in Malaysia found that physicians respond to requests by family because they felt responsible for their family members; they felt they were able to treat family members better than other physicians, providing easy access to hospitals and medications [5].

Additionally, the participants raised concerns that the guidance in the literature for physicians to consider before answering medical requests was not sufficient. It is necessary to empower physicians with tools to assist decision-making, such as the seven steps by La Puma et al. The tool consists of the seven-question method: physicians are required to ask themselves seven important questions before dealing with medical requests from family and friends. These include: Can I undertake an intimate history and exam if needed? Is answering such requests likely to result in intrafamilial issues? Am I willing to be held accountable for any decision I have taken [2,18]? Moreover, other institutes and communities have a clear policy that regulates treating the self, family members, and other close personal relations, which sets a clear path for physicians to act accordingly [19], even though physicians still bypass it sometimes. It is imperative to enforce a policy prohibiting physicians from treating their family members, address existing system inefficiencies, and ensure access to care.

Our study offered a comprehensive insight into the reasons why family physicians choose to treat, or refrain from treating, their own family members and friends. However, its applicability was constrained by the inherent limitations of the qualitative research design, which lacks generalizability. Nevertheless, the exploratory nature of qualitative research played a crucial role in illuminating globally prevalent practices, suggesting that similar contexts could benefit from our findings. Subsequent research could leverage the factors we identified to develop targeted interventions and foster the development of guidelines for physicians. Such efforts could also enhance community awareness about the potential risks and implications associated with receiving medical care from a family member.

## 5. Conclusions

In conclusion, this study shows that treating family members presents several ethical and professional challenges. Policymakers can address this intricate dilemma by improving laws, system regulations, and guidelines for physicians and communities. This will improve access to care, reduce system abuse, empower providers, and enhance community awareness and compliance.

## Figures and Tables

**Figure 1 healthcare-12-02021-f001:**
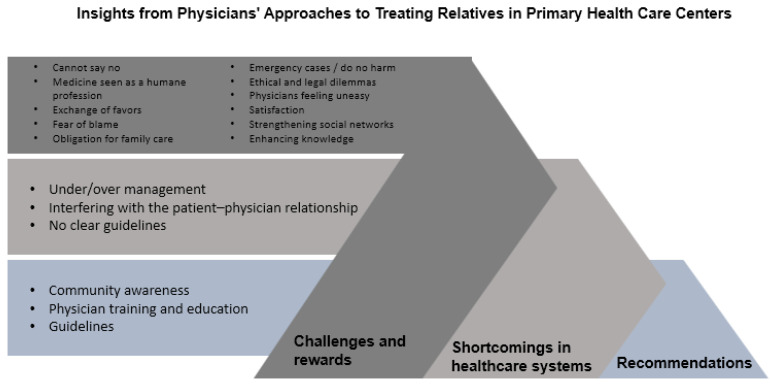
Insights from Physicians’ Approaches to Treating Relatives in Primary Health Care Centers.

**Table 1 healthcare-12-02021-t001:** The participants’ characteristics.

Gender	N (19)
Female	14 (74%)
Male	5 (26%)
Experience (years)	
2–5	13 (68%)
5–10	3 (16%)
>10	3 (16%)
Degree	
Consultant	5 (26%)
Senior Registrar	7 (37%)
Resident	7 (37%)
Nationality	
Saudi	17 (89%)
Non-Saudi	2 (11%)

## Data Availability

The data from the interviews were kept on a locked, password-protected computer accessed only by the primary researcher. The data will be stored for up to five years after publication to enable the verification of data if challenged. To ensure participant confidentiality, all identifying information has been anonymized. The data will be made available upon reasonable request to qualified researchers, provided they agree to maintain the confidentiality of the participants and use the data solely for academic purposes. After five years, all documents will be deleted and disposed of appropriately in compliance with institutional and national data management guidelines.

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
