# Peer review of "Examining Physicians’ Approaches to Treating Relatives in Primary Health Care Centers: Insights from a Qualitative Study"

_healthcare, 2024, doi:10.3390/healthcare12202021_

Round 1
Reviewer 1 Report
Comments and Suggestions for Authors
I read with interest the article by Alhamdan et al. This article may have implications for the country in which it was conducted. It is a relatively culture-based topic and differs in every culture depending on societal norms.
The abstract is very comprehensive but long. Please revise it to be more engaging, reader-friendly, and concise.
Please add the exact questions you asked participants to the methods section or supplementary file.
I kindly recommend that the authors use one of the available checklists for standard reporting of qualitative studies, e.g., COREQ, and include it as a supplementary file for review.
I could not find Appendix 1.
Author Response
- I read with interest the article by Alhamdan et al. This article may have implications for the country in which it was conducted. It is a relatively culture-based topic and differs in every culture depending on societal norms.
- The abstract is very comprehensive but long. Please revise it to be more engaging, reader-friendly, and concise.
Response
Thanks for your comments, The abstract was shortened as advised
- Please add the exact questions you asked participants to the methods section or supplementary file.
Response
Thanks for your comments, it was added to the manuscript after the references as appendix 1
- I kindly recommend that the authors use one of the available checklists for standard reporting of qualitative studies, e.g., COREQ, and include it as a supplementary file for review.
Response
Thanks for your comments, it was added to the manuscript after the references as appendix 2
- I could not find Appendix 1.
Response
Thanks for the comments. It was submitted as the supplementary materials.
Reviewer 2 Report
Comments and Suggestions for Authors
The work submitted for review shows us the dilemmas of family doctors in Saudi Arabia who treat their loved ones. Primary medical care services for people related by family are provided all over the world. This is due to easy access, quick solution in availability and possibility of fulfilling the request.
According to the reviewer, the work requires correction and supplementation of information:
1. Did the interviews contain standard, repetitive questions?
2. Did they contain key words that guided respondents to search for answers to the research hypothesis?
3. Were key words used in the transcription of the text, or was it based on the statements of the respondents and their opinions and assessments?
4. The study is very general and does not allow for drawing specific conclusions, the solutions of which should be sought.
5. Specific problems - What? line 180, how many respondents did they concern?
6. The study does not present the difficulties defined by the respondents in the family doctor relationship, only the social aspect is described.
7. The discussion is based only on our own results without reference to other researchers. It does not contain information or solutions that others have presented, which could be a solution in your "own backyard"
Author Response
Reviewer 2
The work submitted for review shows us the dilemmas of family doctors in Saudi Arabia who treat their loved ones. Primary medical care services for people related by family are provided all over the world. This is due to easy access, quick solution in availability and possibility of fulfilling the request.
According to the reviewer, the work requires correction and supplementation of information:
- Did the interviews contain standard, repetitive questions?
Response:
Yes, the interviews were guided by a topic guide that was created by the author (NA) based on previous literature [4,5,15,16]. This topic guide was reviewed by two physicians to ensure its understanding and relevance to Saudi culture. While the topic guide served as general guidance and included standard questions, additional probes were used as needed to motivate physicians to express their points of view more fully. The topic guide, including the standard questions, is presented in Appendix 1.
- Did they contain key words that guided respondents to search for answers to the research hypothesis?
Response:
The topic guide was developed to cover a broad range of relevant themes based on previous literature, ensuring comprehensive coverage of the subject matter. While the questions were designed to be open-ended to allow for in-depth exploration of the respondents' views, they did include key words related to the research hypothesis to ensure that the discussions remained focused on the core topics of interest. This approach helped to guide respondents in discussing areas pertinent to the research hypothesis without leading their responses. It is important to note that in qualitative research, the primary objective is not to test or prove a hypothesis, but rather to understand and explore the topic in depth. The purpose of the interviews was to gain rich, detailed insights into the perspectives and experiences of the participants, rather than to confirm predefined hypotheses. Therefore, the inclusion of key words was intended to facilitate discussion around relevant themes rather than to direct respondents towards specific answers.
- 3. Were key words used in the transcription of the text, or was it based on the statements of the respondents and their opinions and assessments?
Response:
In the transcription and analysis of the interview data, the data was anonymized then the transcripts were thematically analyzed using MAXQDA Analytics Pro 2020 (VERBI Software). Thematic analysis is a qualitative research method used to identify, analyze, and interpret patterns or themes within a dataset. This method involves systematically organizing and categorizing data to uncover meaningful insights and understand the underlying meanings and experiences of the participants.
In response to the specific question, the transcription was based on the statements of the respondents and their opinions and assessments. While the thematic analysis involved identifying key words and patterns within the data, the initial transcription process aimed to capture the full context and nuances of the respondents' views verbatim. This approach ensured that the richness and depth of the participants' experiences were preserved for a comprehensive analysis.
- 4. The study is very general and does not allow for drawing specific conclusions, the solutions of which should be sought.
Response:
Thank you for your insightful comments. The primary aim of qualitative studies, including this one, is not solely to draw definitive conclusions but to provide a comprehensive and nuanced understanding of the topic under investigation. By capturing the full statements, opinions, and assessments of the respondents verbatim, our approach ensures that we preserve the richness and depth of their experiences and perspectives. The thematic analysis method used in this study allowed us to systematically identify, analyze, and interpret patterns within the dataset, facilitating a deeper exploration of the underlying meanings and experiences of the participants. This approach helps to uncover meaningful insights that might not be immediately apparent through quantitative methods alone. Our goal was to provide a well-rounded and detailed view of the topic, reflecting the complexity and diversity of the participants' views. This comprehensive understanding can inform future research, policy-making, and practice in a more meaningful way
- 5. Specific problems - What? line 180, how many respondents did they concern?
Response:
Thanks for the comments but I don’t understand the point.
- The study does not present the difficulties defined by the respondents in the family doctor relationship, only the social aspect is described.
Response:
Thank you for your comments. We would like to clarify that the study does indeed present the difficulties defined by the respondents in the family doctor relationship, beyond just the social aspects. Our data analysis identified three main themes: perceived benefits and costs of cultural and social connectedness, shortcomings in patient management and healthcare systems, and recommendations to address challenges between physicians and patients who are relatives. While the first theme addresses the social aspects and highlights the advantages and disadvantages of the cultural and social ties between physicians and patients who are relatives, the second theme explores the challenges and difficulties related to patient management and the broader healthcare system, which are critical factors affecting the physician-patient relationship. The third theme provides practical suggestions and solutions offered by the respondents to improve the relationship and address the identified challenges. While the social aspect is indeed one of the main findings, our study also emphasizes other significant factors that impact the family doctor relationship. We believe that this comprehensive view provides a deeper understanding of the complexities involved.
- The discussion is based only on our own results without reference to other researchers. It does not contain information or solutions that others have presented, which could be a solution in your "own backyard"
Response:
Thank you for your valuable feedback on our discussion section. We have expanded the discussion to include a critical analysis that compares and contrasts our findings with other relevant studies, particularly focusing on the context of Saudi Arabia, the GCC region, and global perspectives.
Reviewer 3 Report
Comments and Suggestions for Authors
Many thanks for providing an interesting research manuscript on an interesting subject. After reviewing the paper, I have outlined some critical comments and suggestions to enhance its clarity, depth of analysis, and overall impact. I hope these suggestions will help strengthen your manuscript and its contribution to the field.
- The introduction provides context however I cannot find any local relevance to the GCC nations or global relevance to the “Examining Physicians' Approaches to Treating Relatives in Primary Health Care Centers” which can strengthen the manuscript. Please include a section or a paragraph on how other healthcare systems in other countries handle these ethical issues so that a broader perspective is provided.
- Please use COREQ (Consolidated criteria for reporting qualitative research) checklist guidelines and checklist to report your findings and please attach COREQ in appendices.
- http://intqhc.oxfordjournals.org/content/19/6/349.long
- https://www.cmajopen.ca/content/cmajo/suppl/2016/04/29/4.2.E200.DC1/open-2015-0086-coreq-checklist.pdf
- With regards to the study design, it is not clear whether the interview guide was pilot tested and the details of the feedback we received and how it was incorporated. I cannot see any attachment of the interview guide please try to attach the interview guide for review as an appendix. I am unable to see the attached topic guide.
- With regards to sampling, I am unable to see how non responders were handled. have you considered any response rate if so, please add details of the response rate and how participants were managed if they did not respond to the invitation
- The inclusion criteria are not very clear. Please provide a clear criterion and inform the rationale of this choice.
- Who was the primary researcher and how did you handle interviewers’ characteristics or potential biases which can have an influence of the response of the participants
- Was there check on any intercoder reliability using the software. If so, please add Cohen’s kappa
- The discussion needs further work to enhance the results as it lacks in depth interpretation of results considering the existing literature. Though I can see that the results are stated, there is limited critical analysis that collates the findings to the broader theoretical and practical context. Please compare and contrast your results with those of other studies similar in context and emphasizing how your findings align or differ to the existing literature especially to Saudi Arabia or to the local GCC or Global.
- Please provide more specific and actionable policy recommendation on how primary care institutes can implement policies based upon the findings in the study.
Comments on the Quality of English Language
Moderate editing of English language required.
Author Response
Reviewer 3
Many thanks for providing an interesting research manuscript on an interesting subject. After reviewing the paper, I have outlined some critical comments and suggestions to enhance its clarity, depth of analysis, and overall impact. I hope these suggestions will help strengthen your manuscript and its contribution to the field.
Response:
Thanks for your efforts
- The introduction provides context however I cannot find any local relevance to the GCC nations or global relevance to the “Examining Physicians' Approaches to Treating Relatives in Primary Health Care Centers” which can strengthen the manuscript. Please include a section or a paragraph on how other healthcare systems in other countries handle these ethical issues so that a broader perspective is provided.
Response
Thank you for your valuable feedback. We appreciate your suggestion to include a broader perspective on how other healthcare systems handle the ethical issues related to physicians treating their relatives, which will strengthen the manuscript. We have expanded introduction that incorporates local relevance to the GCC nations and global relevance
- Please use COREQ (Consolidated criteria for reporting qualitative research) checklist guidelines and checklist to report your findings and please attach COREQ in appendices.
- http://intqhc.oxfordjournals.org/content/19/6/349.long
https://www.cmajopen.ca/content/cmajo/suppl/2016/04/29/4.2.E200.DC1/open-2015-0086-coreq-checklist.pdf
Response
Thanks, or your comments. Added.
- With regards to the study design, it is not clear whether the interview guide was pilot tested and the details of the feedback we received and how it was incorporated. I cannot see any attachment of the interview guide please try to attach the interview guide for review as an appendix. I am unable to see the attached topic guide.
Response
The topic guide was meticulously created by the author (NA) based on a comprehensive review of previous literature to ensure its appropriateness and relevance within the Saudi cultural context, the topic guide was reviewed by two experienced physicians. Their feedback was instrumental in refining the guide to ensure clarity, cultural sensitivity, and relevance to the local healthcare environment. While the topic guide was not pilot tested in a formal sense as in quantitative design, the iterative feedback process with the two physicians served a similar purpose. Their insights helped us identify potential areas for improvement, which were then incorporated into the final version of the guide. This iterative review process ensured that the questions were understandable, culturally appropriate, and capable of eliciting rich, detailed responses from the participants. The topic guide served as a general framework during the interviews, allowing for flexibility and the use of additional probes to motivate physicians to express their points of view more comprehensively. This approach enabled us to capture a wide range of perspectives and experiences, enhancing the depth and quality of the data collected. We have included the topic guide as an appendix (Appendix 1).
- With regards to sampling, I am unable to see how non responders were handled. have you considered any response rate if so, please add details of the response rate and how participants were managed if they did not respond to the invitation
Response
Thank you for raising this important point. We appreciate the opportunity to clarify our sampling strategy and address the reviewer's concerns about non-responders and response rates. In qualitative research, the focus is on gaining in-depth understanding and rich insights from a smaller, carefully selected sample, rather than on achieving statistical representativeness as in quantitative studies. Therefore, the concept of response rate, which is crucial in quantitative research, is not as central in qualitative research. Our study employed purposive sampling, a common approach in qualitative research, to select participants who could provide valuable insights based on specific criteria. We targeted junior and senior family medicine physicians working for the Ministry of Health in Riyadh, Saudi Arabia, as they represent a key stakeholder group in the context of our research. Invitations for participation were emailed to the primary healthcare centers, and interested participants who met the inclusion criteria were contacted to arrange interviews. While we did not formally track the response rate, we continued to recruit participants until data saturation was achieved, which is a common practice in qualitative research. This means that we continued interviewing participants until no new information or themes emerged, indicating that we had captured a comprehensive range of perspectives on the topic.
- The inclusion criteria are not very clear. Please provide a clear criterion and inform the rationale of this choice.
Response
Thank you for pointing out the need for clearer articulation of the inclusion criteria. We appreciate the opportunity to provide a more detailed explanation of our participant selection process. It was added to the manuscript.
- Who was the primary researcher and how did you handle interviewers’ characteristics or potential biases which can have an influence of the response of the participants
Response
Both Alhamdan and Aloudah contributed equally to this work as primary investigators, Aloudah’ s previous work, which can be found on Google Scholar under the name Nouf Aloudah, includes numerous published studies that address similar contexts and themes. This background has equipped her with the expertise to handle qualitative data with the necessary rigor and depth. In qualitative research, our focus is on understanding the depth and complexity of human experiences, which often does not lend itself to the same quantitative reliability checks and biases used in quantitative research. Specifically, we did not use Cohen’s kappa to measure intercoder reliability. Instead, we employed rigorous qualitative methods to ensure the credibility and trustworthiness of our findings, in line with Guba and Lincoln’s criteria for qualitative research, which included the following: we used multiple data sources (focus groups and face-to-face interviews) to capture a comprehensive view of the phenomena under study, thereby enhancing the richness and depth of our data, member checking: after the initial analysis, we shared our findings with a subset of participants to ensure that our interpretations accurately reflected their experiences and perspectives. Further, peer debriefing where we engaged in regular debriefing sessions with fellow researchers to discuss the coding process and emerging themes, which helped to identify and address any potential biases. Furthermore, we maintained a comprehensive audit trail that documented our methodological decisions, data collection, and analysis processes, ensuring transparency and dependability. By employing these strategies, which was documented in the data analysis section, we aimed to ensure the rigor and trustworthiness of our qualitative study.
- Was there check on any intercoder reliability using the software. If so, please add Cohen’s kappa
Response
Thank you for your insightful feedback regarding the methods used in our study. We appreciate your suggestion to include a check on intercoder reliability using Cohen’s kappa. However, we would like to clarify our approach within the context of qualitative research.
In qualitative research, our focus is on understanding the depth and complexity of human experiences, which often does not lend itself to the same quantitative reliability checks used in quantitative research. Specifically, we did not use Cohen’s kappa to measure intercoder reliability. Instead, we employed rigorous qualitative methods to ensure the credibility and trustworthiness of our findings, in line with Guba and Lincoln’s criteria for qualitative research, which included the following : we used multiple data sources (focus groups and face-to-face interviews) to capture a comprehensive view of the phenomena under study, thereby enhancing the richness and depth of our data, member checking: after the initial analysis, we shared our findings with a subset of participants to ensure that our interpretations accurately reflected their experiences and perspectives. Further, peer debriefing where we engaged in regular debriefing sessions with fellow researchers to discuss the coding process and emerging themes, which helped to identify and address any potential biases. Furthermore, we maintained a comprehensive audit trail that documented our methodological decisions, data collection, and analysis processes, ensuring transparency and dependability. By employing these strategies, which was documented in the data analysis section, we aimed to ensure the rigor and trustworthiness of our qualitative study. We believe that these approaches are more appropriate for our research design and better suited to capturing the nuanced and complex nature of the data. My previous work, which can be found on Google Scholar under the name Nouf Aloudah, includes numerous published studies that address similar contexts and themes. This background has equipped me with the expertise to handle qualitative data with the necessary rigor and depth.
- The discussion needs further work to enhance the results as it lacks in depth interpretation of results considering the existing literature. Though I can see that the results are stated, there is limited critical analysis that collates the findings to the broader theoretical and practical context. Please compare and contrast your results with those of other studies similar in context and emphasizing how your findings align or differ to the existing literature especially to Saudi Arabia or to the local GCC or Global.
Response
Thank you for your valuable feedback on our discussion section. We have expanded the discussion to include a critical analysis that compares and contrasts our findings with other relevant studies, particularly focusing on the context of Saudi Arabia, the GCC region, and global perspectives.
- Please provide more specific and actionable policy recommendation on how primary care institutes can implement policies based upon the findings in the study.
Response
Thank you for your insightful comments and for highlighting the need for more specific and actionable policy recommendations, it was added to the discussion.
We believe that these revisions have addressed all the concerns raised, and we hope that the manuscript now meets the required standards for publication. Thank you again for your valuable feedback.
Sincerely,
Nouf Aloudah
Round 2
Reviewer 1 Report
Comments and Suggestions for Authors
Thank you for the revisions. I have no further comments.
Author Response
Dear Reviewer
Thank you very much for your kind words and for accepting our responses. We are thrilled to hear that you have no further comments and appreciate your thoughtful review and feedback throughout the process. Your insights have contributed to enhancing the quality of our manuscript, and we are grateful for your support and guidance.
We are excited to see our work published and hope it will make a valuable contribution to the field. Thank you once again for your time and effort in reviewing our paper.
Nouf Aloudah
Reviewer 2 Report
Comments and Suggestions for Authors
The additions and clarifications introduced are sufficient
Author Response

(The authors gave the same response as above.)

Reviewer 3 Report
Comments and Suggestions for Authors
Many thanks for addressing the peer- review comments positively.
Author Response

(The authors gave the same response as above.)
